# Mycofabrication of Mycelium-Based Leather from Brown-Rot Fungi

**DOI:** 10.3390/jof8030317

**Published:** 2022-03-19

**Authors:** Jegadeesh Raman, Da-Song Kim, Hyun-Seok Kim, Deuk-Sil Oh, Hyun-Jae Shin

**Affiliations:** 1Department of Chemical Engineering, Graduate School of Chosun University, 309 Pilmun-daero, Dong-gu, Gwangju 61452, Gyeonggi, Korea; jegadeesh_ooty@rediffmail.com (J.R.); dasong1214@daum.net (D.-S.K.); 2Agricultural Microbiology Division, National Institute of Agricultural Science, Rural Development Administration, Wanju-Gun 55365, Jeollabuk-do, Korea; 3Jeollanam-do Forest Resources Research Institute, 7 Dado-ro, Naju-8213, Sanpo-myeon, Naju 58213, Jeollanam-do, Korea; shitake@korea.kr (H.-S.K.); ohye@korea.kr (D.-S.O.)

**Keywords:** mushroom, mycelium, leather, composite, Polyporales, PEG, plasticizing, tensile strength

## Abstract

Sustainable substitutes for leather can be made from mushroom mycelium, which is an environmentally friendly alternative to animal and synthetic leather. Mycelium-based leather is derived from Polyporales, in which lignocellulosic material is used as the substrate. The plasticizing and crosslinking of mycelial mats with various reagents might affect the leather properties and mycelial architecture. This study investigated the physicochemical and mechanical properties of mycelium-based leather (MBL) samples, including the hygroscopic nature, thermal stability, cell wall chemistry, density, micromorphology, tensile strength, elongation rate, and Young’s modulus. Micromorphological observations confirmed the mycelial networks and their binding performance, verifying their efficacy as a substitute leather. The most significant effects were observed after treatment with 20% polyethylene glycol, which resulted in an increase in Young’s modulus and tensile strength. Furthermore, the samples generally exhibited a high density (1.35, 1.46 g/cm^3^) and tensile strength (7.21 ± 0.93, 8.49 ± 0.90 MPa), resembling leather. The tear strength reached as low as 0.5–0.8 N/mm. However, the tensile and tear strength may be affected by leather processing and the tuning of mycelial growth. Nevertheless, high-density mycelia are shown to be suitable for the production of MBL, while mycofabrication and strain selection are sustainable for novel industrial applications of MBL.

## 1. Introduction

Mushroom-based leather substitutes are an emerging green and sustainable class that can replace various animal and synthetic leather products, to reduce intolerable environmental stress. However, mycofabrication and mushroom-based leather production have not yet been explored. Mycelium is the vegetative part of a mushroom consisting of a mass of branching, fibrous, and natural composite materials with controlled physio-mechanical properties. After chemical and hot treatments, it becomes extremely durable and resistant to environmental stress [1]. Thus, mycelium applications have enormous potential to benefit both humans and the planet. Upcycling low-cost agro-waste substrates yields mycelium-based leather (MBL) with added supplements. Fungi and plant materials are the most suitable combinations for future novel biomaterial development. This new strategy can reduce the health and environmental risks associated with the production of bovine leather and is an alternative to petroleum-based polymeric foam. It has become a highlight in biomaterial engineering, owing to its zero pollution and renewability during the formation and treatment processes [2]. In recent years, mushroom mycelia have shown favorable characteristics for developing sustainable biomaterials. Such types of mycelia include mycelial biocomposites [3,4], mushroom leather [5], foams [6], and mycoboards [7]. Mushroom mycelia are whitish brown, leathery, resistant to puncture, and show different physical and mechanical characteristics [8]. Basidiomycete mycelia are versatile, with dendritic patterns of branching networks and autonomous hyphae. Fungal mycelia are inconspicuous and primarily embedded in the substrate/host. In addition, these mycelia are hardened, stored, and pigmented.

Most mushroom species are saprophytic; they bind to and digest lignocellulose, developing dense mycelia. Solid-state fermentation degrades agro-industrial waste substrates, such as agricultural and industrial residues. They mainly consist of cellulose, hemicellulose, and lignin and are collectively defined as lignocellulosic materials [9]. Polyporales are major wood-decay poroid fungi belonging to the cosmopolitan Basidiomycota genus. Most Polyporales species bio-convert agro-residue and forest waste into other resources without harming or depleting the natural ecosystem. In addition, Polyporales mycelia penetrate substrates and can fill their volume with a very dense mycelial network, enabling the formation of mycelium-based biocomposites. Polyporales species are suitable for mushroom-based biomaterial production [10]. Their fruiting bodies vary from soft to tough because of their hyphal structures and cell-wall chemistry. The mass of mushroom hyphae is vegetative, and the reproductive stage can be more conspicuous. Their cell walls are composed of matrix components, which are flexible structures composed of chitin, glucans, and glycoproteins. The cell-wall polymer architecture and chemical composition (glucan and chitin) greatly influence external ecological factors [11]. In addition, the physio-mechanical properties of mycelia are affected by the substrate type, incubation conditions, and fabrication process. However, selecting suitable strains and conditions for mycelium-based composites and leather production is complex. A monokaryotic culture is typically the preferred option; however, hybridization and mutation may improve the strength, flexibility, and durability of MBL.

The commercial potential of MBL has been recognized by both startups and established companies. Leading leather manufacturers include MycoWork (mycoworks.com accessed on 4 December 2021, Bolt Threads (boltthreads.com, accessed on 4 December 2021), Desserto (desserto.com.mz accessed on 4 December 2021), Mycotech Lab (mycl.bio), and Mycel (mycelproject.com accessed on 4 December 2021). Based on their strong financial structure and high-quality personnel, these companies collaborate with leading luxury companies, including Hermes and Mercedes-Benz, to produce basic raw leather, personal luxury items, and car interior materials. The MBL technology can also be used to produce industrial and construction materials. Many aspects of MBL manufacturing are secured by patents or are proprietary. Further development of this field requires freely accessible data from studies ranging from mycology to engineering. For this purpose, the present study provided research data from mushroom selection to MBL production. High-value MBLs were developed from *Fomitella fraxinea* (ASTI 17001) with minimal chemical usage. Besides, MBLs were obtained through the upcycling of low-cost agro-waste substrates. Hotpress, plasticizing, and crosslink methods might alter the mycelial properties and architecture [12]. On the other hand, animal-based leather is made more flexible and durable by chemical treatment.

Interestingly, mycelial-based leather is a vegan-friendly material that replaces various synthetic leathers and animal leather products to reduce intolerable environmental stress. Mushroom MBLs exhibit mechanical and tactile properties and functions comparable to existing industrial products and applications. The new strategy reduces the health and environmental risks associated with the production of bovine leather and presents an alternative to petroleum-based polymeric foam.

## 2. Materials and Methods

### 2.1. Strains Applied in the Study

The strains used in this study were conserved in Lab No. 5105, Engineering Building 2, Chosun University 309, Pilmun-daero, Dong-gu, Gwangju, Republic of Korea. *Fomitella fraxinea* (ASTI 17001) [13], *Ganoderma lucidum* (JF 17-01), *Elfvingia applanata* (JF 23-01), *Bjerkandera adusta* (JF 78-01), *Microporus affinis* (JF 46-01), *Trametes versicolor* (JF 52-01), *Fomitopsis pinicola* (JF 79-01), *Wolfiporia extensa* (JF 47-01), and *Postia balsamea* (JF 80-01) were obtained from Jeollanam-do Forest Resources Research Institute, Naju City, Republic of Korea. *Ganoderma applanatum* (KMCC 02967) was obtained from the Mushroom Research Division, National Institute of Horticultural and Herbal Science, RDA, Republic of Korea. *Formitopsis pinicola* (KCTC 6208), *F. rosea* (KCTC 26226), *Trametes suaveolens* (KCTC 26205), and *T. hirsuta* (KCTC 2620) were purchased from the Korean Collection for Type Cultures, Republic of Korea. All 14 Polyporales species were subcultured and maintained on PDA (Difco, Spark, MD, USA) and yeast, malt, and peptone extract agar (YMPA; containing 10 g L^−1^ of dextrose, 5 g L^−1^ of peptone, yeast, and malt extract at 3 g L^−1^) at 28 °C in the absence of light. After growth, they were stored at 4 °C.

### 2.2. Culture Media and Growth Optimization

For efficient mycelial growth, we used YMPA media (yeast extract (3 g), malt extract (3 g), peptone (5 g), dextrose (10 g), and agar), pH 6, and an incubation temperature of 28 °C for all Polyporales. YMPA liquid/solid medium is the optimum culture medium for culture maintenance, liquid spawn production, and mycelial mat production. The medium was sterilized at 15 psi and 121 °C for 20 min and dispensed in a sterile Petri plate. Polyporale cultures are routinely grown in YMPA. In each experimental treatment, three replicate plates were inoculated on the circumferential edge of the media using 5-mm diameter plugs taken from the periphery of exponentially growing cultures of each strain grown on YMPA. Inoculated culture plates were sealed and incubated at 28 °C. Radial growth was determined by measuring the colony radius minus the diameter of the inoculum 90-mm petri dish on four perpendicular axes [14]. The diameter of mycelial growth was measured (days one to four), and the mycelial density was determined qualitatively, classified as very thick (++++), thick (+++), thin (++), or very thin (+) (Appendix A). The incubation period was defined as the number of days from inoculation to complete mycelial ramification (1–4 d), and the results were analyzed using ANOVA (SPSS, version 25, SPSS Inc., Chicago, IL, USA). The liquid medium (250 mL) culture density was calculated from the fresh weight and dry weight (g) of the harvested mycelium.

### 2.3. Mycelial Linear Growth Rate

The test tube experiment determined linear growth measurements. The substrate was prepared with sawdust and rice bran (8:2) in glass test tubes and sterilized at 121 °C for 60 min. After the inoculation process was conducted, the mycelium growth rate was determined at 24-h intervals (mm/day). Mycelium growth was measured in millimeters using a ruler until the entire distance covered by the hyphae (linear length) was measured. All experiments were conducted in triplicate.

### 2.4. Culture Media and Spawn Production 

Liquid spawns were prepared using YMPB (200 mL) in 500 conical flasks under controlled conditions. On the cellophane membrane surface (overlaid onto YMPA), 14 Polyporale mycelia were grown for 6 d. The grown mycelium was transferred to a liquid medium and fragmented with a Waring blender for 20 s at low speed (Nippon Seiki Co., Ltd., Niigata, Japan). The fragmented mycelia were transferred to identical conical flasks and incubated for 3 d under stationary conditions and for 4 d under shaking conditions (agitation speed, 150 rpm). The medium containing the mycelia was re-homogenized, and 100 mL of fresh YMPB medium was added. The culture flasks were incubated for 7 d under shaking conditions.

### 2.5. Box Cultivation and Substrate Preparation

The cultivation substrate (2 kg) was prepared using oak sawdust and rice bran (8:2) in polypropylene bags. A 0.75-inch diameter hole was bored in the center of the substrate and sterilized by autoclaving at 121 °C for 40 min. Then, 50 mL of 14-day-old liquid spawns was inoculated in the center of the substrate and incubated in a dark room for 18 d at 28 °C and 80–90% humidity for a spawn run. The mycelial-grown substrate was transferred to boxes (155 × 155 × 87 mm^3^, HPL822D, LOCK & LOCK, Republic of Korea). The box lids were holed evenly and covered with synthetic 20 mm × 0.3 μm filter paper stickers (filter disc mushroom applied under wide mouth jar lid for mushroom cultivation (256), MS1644, China). The substrate was transferred to boxes for mycelial mat production. Simultaneously, the box-filled substrate was hand-pressed for uniform distribution and allowed to grow further at 28 °C and 80–90% humidity at a high CO_2_ level (1500 ppm). After 3 d, the substrate was overlaid with grade 10 cheesecloth (20 × 12 threads per square inch). The experiment consisted of six boxes for each strain, and the box cultures were incubated for 40 d in the dark.

### 2.6. Mycelial Harvesting and Mycofabrication

The mycelial mat was harvested by peeling, and the fresh weight and size were measured immediately. The density was calculated using the weight after drying, and the volume and yield of each specimen were measured based on the substrate weight. For plasticization, the harvested mycelial mats were soaked in 15% glycerol, 15% ethylene glycol, and 20% polyethylene glycol (PEG-M_n_ 400, Sigma-Aldrich Korea) for 48 h, each separately. Coating and crosslinking were performed with 20% corn zein and 5% tannic acid after plasticizing. The mycelial mats were drained, dried at room temperature, and rolled evenly.

### 2.7. Specimen Preparation and Analysis

Mycelial mats were cut from 18 different treatment samples by manual vertical sawing. Tiny blisters on the samples were removed manually using a press roller. They were cut into a rectangular shape for hotpress treatment and tensile tests (15 × 20 mm^2^), or into square specimens for density testing (5 × 5 mm^2^), SEM and FTIR (2 × 2 cm^2^), and hydrodynamic characterization (3 × 3 cm^2^). Hotpress treatment (60 and 120 °C) was performed on standard bench manual presses for 20 min. The mycelial mat exposed to hotpressing was cooled to room temperature, whereas the cool-pressed mat was dried for 48 h.

### 2.8. Physical and Mechanical Properties Analysis

The specimen dimensions (size 15 × 20 mm^2^ and sample diameter measured by Vernier caliper scale) were measured before testing by a Universal Testing Machine (UTM), and 10 specimens of each material were tested for tensile strength and elongation. Tests were performed using a Zwick/Roell Z010 UTM (Ulm, Germany) at an elongation rate of 2 mm/min and maximum force of 1 kN. Data were analyzed to obtain stress-strain plots, the tensile strength, and Young’s modulus. The density was determined using a gas pycnometer (Accupyc II 1340/Micromeritics, Norcross, Georgia, USA). The surface morphology of the dried samples was examined using field-emission scanning electron microscopy (FE-SEM, Quanta 450, EDX-OXFORD) operated at 5–10 kV.

### 2.9. Chemical Properties Analysis

The chemical composition of the MBL sample was analyzed using FT-IR spectroscopy (Nicolet 6700, Thermo Scientific, USA) in the range of 4000–6000 cm^−1^ with a resolution of 1 cm^−^^1^. Cell-wall chemistry was measured using high-performance anion-exchange chromatography (HPAEC, ICS-5000, Dionex Co., Sunnyvale, California, USA). Elemental analyses of the MBL samples (C-control, CH-control hotpress, PEG-C, and PEG-H) were performed using an energy-dispersive X-ray spectroscope attached to a field emission scanning electron microscope. Thermogravimetric analysis (TGA) was performed using a TGA Q50 instrument (TA Instruments, New Castle, DE, USA). Measurements were performed with biological duplicates of 25 mg of mycelium in a platinum pan using a 100 mL/min airflow. The temperature increased from 20 to 600 °C at a rate of 10 °C/min. All experimental data are presented as the mean ± standard deviation, and statistical analyses were performed using the SPSS statistical package.

## 3. Results

### 3.1. Culture Media and Growth Optimization of Polyporales Species

Commercial YMPA is a suitable medium for all 14 Polyporales species. Among them, G. applanatum and B. adusta showed fast radial growth at 41.42 mm ± 0.13 mm and 41.00 mm ± 0.39 mm, respectively. Elfvingia applanatum showed a slow growth rate (11.75 mm ± 0.77 mm) compared with other Polyporales. However, the average mycelium diameter ranged from 11.75 mm ± 0.77 mm to 41.42 mm ± 0.13 mm at 4 d after inoculation (Table 1, Appendix A). More interestingly, G. applanatum showed fast mycelial colonization on sawdust substrate, and the average linear growth rate (80.88 mm/16 d ± 7.52 mm/16 d) was comparatively higher than that of other Polyporales species (Table 1). In contrast, the highest liquid culture mycelial density (fresh and dry weights in grams, respectively) was recorded for T. hirsute with 11.33 mm ± 0.17 mm and 1.36 mm ± 0.02 mm, and for T. versicolor with 10.92 mm ± 0.28 mm and 1.33 mm ± 0.51 mm. For comparison, the smallest mycelial weights were observed in F. fraxinea with 4.07 mm ± 0.34 mm and 0.81 ± 0.02 mm, and in W. extensa with 4.33 mm ± 0.76 mm and 0.91 mm ± 0.08 mm, respectively (Table 1 and Appendix A). Liquid cultures of harvested mycelia showed different surface morphology characteristics and mycelial densities. For comparison, all species produced dense mycelia, except *E. applanatum* and *T. hirsuta* (Figure 1). Highly dense and flexible mycelium mats were obtained from *Ganoderma lucidum*, *G. applanatum*, *Fomitella fraxinea*, *F. pinicola*-KCTC, and *Postia balsamea*. In addition, *Formitopsis pinicola*-JF, *F. rosea*, *Trametes versicolor*, *T. suaveolens*, *Wolfiporia extensa*, *Microporus affinis*, and *Bjerkandera adusta* produced high-density mycelia. However, mycelia are not flexible, are highly brittle, and have low physical strength.

### 3.2. Linear Growth Measurement

In this study, sawdust and rice bran (8:2) were used in the measurements of linear growth and mycelial mat production (Figure 2). Among the test Polyporales species, *F. fraxinea* produced a highly dense mycelial mat, with an average linear growth rate of 42.77 ± 0.78. However, a fast growth rate was observed in *G. applanatum*, and slow growth was recorded in *T. rosea* at 80.88 ± 7.52 and 28.10 ± 1.13, respectively (Figure 1 and Appendix A).

### 3.3. Spawn Production from Polyporales Species

This study aimed to investigate the appropriate conditions, growth rates, and nutritional contents of Polyporales species. We optimized different commercial media for liquid spawn production, among which YMPB (liquid broth) is considered the optimum medium for mycelial growth. YMPB showed rapid mycelial growth with a dense colony morphology. The optimum conditions for liquid spawn production were as follows: YMPB medium with an initial pH of 5.5, incubated at 28 °C with an agitation speed of 200 rpm. After 14 d of incubation, the maximum amounts of fresh and dry biomass harvested from T. hirsuta and T. versicolor were 11.33 g/300 mL ± 0.17 g/300 mL and 10.92 g/300 mL ± 0.28 g/300 mL, respectively (Table 1). Mycelium colonization on sawdust rice bran substrate (2 kg) was recorded at 12–18 d intervals. Interestingly, homogeneous and high-density mycelial colonies with uniform distribution were obtained through the liquid spawn (Appendix A).

### 3.4. Box Cultivation and Mycelial Mat Harvesting

In this study, we developed MBL using solid-state fermentation methods. The liquid-spawn-inoculated solid substrates were transferred to a box for mycelial mat production. Alternatively, plastic bags, bottles, and beds can be prepared using polypropylene (PP) boxes. Boxes filled with spawn substrate were incubated at a temperature of 28 °C with a relative humidity of 85% for the next 30–40 d. After 30 d of incubation, primordial initiation was observed in *G. lucidum*, *G. applanatum*, and *W. extensa*. In addition, *E. applanata*, *F. rosea*, *T. hirsuta*, *T. suaveolens*, *M. affinis*, and *B. adusta* failed to produce mycelial mats. Furthermore, *F. pinicola*-JF, *F. pinicola*-KCTC, *T. versicolor*, and *P. balsamea* produced very thin mycelial mats (Figure 2). Interestingly, the *F. fraxinea* culture produced a highly dense mycelial mat. The harvested mycelium was brownish-white in color with a high density. Finally, F. fraxinea was selected for mycelia-based leather production on sawdust substrate. The mycelial mat was evenly peeled on a bio-composed sawdust substrate. Herein, over six constructive experiments, the average substrate weight was 446.40 g ± 28.32 g, and the yield was 12.08% ± 2.91%. The fresh and dry weights of the mycelial mat were 53.96 g ± 13.29 g and 23.57 g ± 4.59 g, respectively (Appendix A). The initial moisture content of the mycelial mat was 55.17% ± 7.55%, and the average shrinkage was 0.74%, calculated according to the size dimensions (18.07 mm ± 0.46 mm × 5.77 mm ± 0.23 mm × 22.67 mm ± 0.52 mm) and fresh and dry weights of the mycelial mat.

### 3.5. Mycofabrication and Mycelium-Based Leather (MBL) Production from F. Fraxinea

Fomitella fraxinea produced a highly dense mycelium on a solid substrate for 30 d. Harvested mycelial mats were immediately soaked in different plasticizing solutions and incubated for 48 h. Crosslinking was carried out during plasticizing, and coating was performed after plasticizing with a dried mycelial mat. We used three different plasticizing agents, containing 15% ethylene glycol and 20% PEG, suitable for MBL production. However, plasticization, crosslinking, and surface-coating may alter the surface color of MBLs. In our work, different plasticizing reagents altered the surface morphology and color (Figure 3). Corn zein coating increased yellowness, and 5–10% tannic acid crosslinking altered the color desirably. MBLs treated with 10% tannic acid showed a dark reddish-yellow color (Figure 3g,h).

In addition, corn zein-treated MBLs exhibited a glassy appearance. The cheesecloth-over-layer MBLs also resemble the color of the plasticizer-treated samples. In this study, 20%-PEG-treated MBL samples were thoroughly investigated for their mechanical, physical, and chemical characterization (Appendix A). This is the first study to report the density, tensile strength, elongation percentage, Young’s modulus, stress-strain curves, thermal analysis, and micromorphology (SEM) of the mycelial cell wall (HPAEC), along with FTIR analysis of MBL, using a fully disclosed protocol with F. fraxinea.

### 3.6. Physical and Mechanical Properties of MBLs

Plasticizing, crosslinking, and surface-coating methods may alter the texture and color of MBLs (Figure 3). The mechanical properties of the developed MBLs are important to measure because mycelia are leather materials, which can respond differently to stress and strain (Table 2). Moreover, with and without hotpress treatment, MBL samples are structurally inhomogeneous in terms of cell-wall component distribution and orientation and thickness, which adds a stochastic character to their mechanical response. The measured parameters, such as elongation percentage, stress, and strain, showed significant differences among all samples, considering different plasticizing, coating, crosslinking, and hotpress treatments. In the present study, glycerol-treated samples without hotpressing and at 60 °C exhibited high elongation at 69.74% ± 5.33% and 58.86% ± 5.19%, respectively.

The tensile strength of the cheesecloth increased as the elongation decreased; 20%-PEG-treated and cheesecloth-over-layered (PEGCO) MBLs show high tensile strength, with Young’s moduli of 8.49 MPa ± 0.90 MPa and 8.14 MPa ± 0.88 MPa, respectively. The highest Young’s moduli were recorded in 20% PEG and 20% PEGCO with hotpress (120 °C) at 6.69 MPa ± 0.67 MPa and 8.14 MPa ± 0.88 MPa, respectively. This study highlights the need for complete drying and rolling to improve the elongation percentage and tensile strength. In addition, the coating reduces the water absorption and smooths the surface. Plasticizing and crosslinking are essential steps for altering tensile and elongation strengths. More interestingly, 20%-PEG-treated MBLs showed high physical strength compared with other MBLs. Therefore, 20%-PEG-treated samples were considered for physicochemical characterization. The densities of the control (C), control hotpressed (CH), 20% PEG without hotpress (PEG-C), and PEG with hotpress (PEG-H) samples were 1.58 g/cm^3^, 1.51 g/cm^3^, 1.35 g/cm^3^, and 1.46 g/cm^3^, respectively (Appendix A).

### 3.7. Micromorphology of MBLs from F. Fraxinea

The SEM images show an interconnected network, highlighting the surface features of the MBLs and control samples (Figure 4). The density of MBLs clearly increased in the treated samples. Interestingly, the 20%-PEG-H MBLs showed uniform structures with smooth surfaces. Control samples showed tube- and thread-like hyphae in the top, bottom, and middle views. Short and highly entangled tube-like hyphae were more common on the surface, whereas the middle and bottom compact filaments increased. In addition, the morphology of the control samples remained almost unaltered; hotpress control samples remained intact, and loosely interwoven hyphae appeared. Significant differences were observed among the treated samples. Furthermore, the binding and crosslinkage mechanisms of the mycelium affected its mechanical and thermal properties.

### 3.8. Chemical Properties of MBLs from F. Fraxinea

The cell-wall sugar composition was measured by high-performance anion-exchange chromatography (HPAEC). For HPAEC analysis, the mycelium was decomposed in all samples with sulfuric acid to produce a glucosamine peak, which is a decomposition product of chitin (Appendix A). The minor cell-wall glucan rhamnose peak intensity exhibited differences between the control and treated MBL samples. Chitin is a structural polysaccharide that represents the physical strength of MBL. The structural modification of the cell wall was confirmed by FTIR spectroscopy. Figure 5 shows a comparison of the spectra of the four samples. Of the PEG-treated samples, 20% exhibited highly intensive bands, as compared with the control samples (C, CH). The MBLs of F. fraxinea led to a small decrease in the intensities of carbohydrates in bands C and CH at 991 and 997 cm^−1^. The increase in carbohydrates was most pronounced at 1075.5 and 1064.5 cm^−1^, as recorded in the PEG-treated samples. The spectra also revealed a small peak at 1375 cm^−1^, assigned to chitin. The stretching vibration frequencies of polysaccharides were observed in the range of 1189–1899 cm^−1^.

Finally, the broadband centered at 3400 cm^−1^ was attributed to the hydroxyl groups of carbohydrates, the main component of cell walls in the MBLs. The relationship between chitin, β-glucan, and the mannan percentage was analyzed in the MBLs and control samples, and the β-glucan content was recorded, followed by those of chitin and mannan. The SEM–EDX results for the control and MBL samples are shown in Figure 6. From the results, the control samples C and CH show C weight percentages of 56.3 and 56.32, respectively, with atomic percentages of 64.47 and 64.93, respectively, indicating that a relatively higher percentage of C is present in the control samples than in the plasticized samples (20% PEG). For the plasticized samples, a relatively high amount of C is indicated, with C weight percentages of 60.72 and 60.91 and atomic percentages of 67.50 and 67.71. Oxygen was also observed at the maximum level in all samples, whereas Mg, S, K, and Ca were recorded at low levels.

The thermal stability and water contact angle (CA) of the MBLs, as well as different plasticization and hotpress treatments, were compared to the control (untreated mycelial mat) samples. The biodegradable MBL proved to be thermally stable up to 300 °C (Figure 7a–d). Water contact-angle tests revealed the hygroscopic and hydrophobic nature of the MBL. The CA was measured after the MBLs were conditioned both up and down for 24 h (Figure 7e); MBLs were not found to be stable under either condition. Furthermore, PEG-treated MBLs, PEG-C and PEG-H, exhibited low CA values on both sides—30.84° ± 12.55° and 97.58° ± 7.94° for upside conditioning and 59.87° ± 14.52° and 74.16° ± 20.34° for downside conditioning, respectively. These PEG-treated MBLs were more sensitive to moisture adsorption. Meanwhile, the untreated mycelial mat (C) was less sensitive to moisture absorption, with a CA of 129.63° ± 19.32° for the upside mycelia and 75.47° ± 15.10° for the downside mycelia, respectively. However, the PEG-C samples showed a more hydrophobic nature than PEG-H. In PEG-C samples, the interlayer was incorporated with polyethylene glycol, which has a high affinity for water. The comparison data of the elongation, tensile strength, and Young’s modulus are shown in Figure 7f.

## 4. Discussion

Today, mushrooms are used for food and medicine and in the development of engineering materials [15]. Mycelium-based biomaterials offer an alternative fabrication paradigm based on the growth of materials, rather than their extraction. Agricultural residue fibers are inoculated with fungal mycelia, which form an interwoven three-dimensional filamentous network that binds the feedstock into a lightweight material [16]. Most Polyporales have shiny surfaces of a woody nature, owing to the pigmentation on the surface, and they are highly suitable for the development of mushroom-based biomaterials [10]. They are distributed widely in the forest ecosystem, where they decompose deadwood, recycling major nutrients in the system [17].

Nevertheless, Polyporales are suitable for natural bio-composited production, and the vegetative parts of the fungi can be produced as sustainable alternatives to MBL and construction materials [4,18]. Mushroom-based biomaterials are alternative biodegradable materials and products derived from renewable resources. These natural biomaterials will be replaced with various petroleum-based products to reduce the intolerable stress placed on the planet’s environment. Recently, many studies have shown the potential of developing packaging, building, textile, and transparent edible films using fungi/mushrooms [17,19]. However, few studies have been conducted on MBL production and processing.

In the present study, 14 different Polyporale species were screened for MBL production. During optimized mycelial growth, the Polyporales species showed remarkable radial growth and density differences in solid and liquid media (YMP agar, YMP broth). Song et al. (2016) observed the high mycelial density of *Pleurotus ostreatus* on YMPA media [20]. YMPA was the best carbon and nitrogen source for Polyporales species. Moreover, maltose has been shown to be the best carbon source for mushroom species [21]. Bae et al. (2021) reported that semi-solid and liquid culture media are suitable for high-density, flexible, and high-strength mycelial mat production [18]. In this study, solid and liquid media containing YMP were found to be optimal for all test species. Fast radial growth in *G. applanatum* and *B. adusta* was observed, as compared to other Polyporales species. *F. fraxinea* (ASTI 17001) showed moderate radial growth and very low weights (fresh and dry) in a liquid medium. Interestingly, liquid-cultured mycelia exhibited greater physical strength than the other test strains. However, we obtained high-density mycelia through solid-state fermentation techniques. 

Spawn quality and rapid colonization are among the most critical factors influencing MBL production. The spawn is a fungal seed that holds a specific strain of mushroom mycelia. Besides quality spawn, rapid colonization on substrates without contamination risk is important in industrial applications. Recently, four different spawn types were used for large-scale mushroom cultivation [22,23]. Grains and sawdust spawns are most commonly used because of their ready availability, low cost, and low equipment investment. Meanwhile, they cause a high contamination rate owing to heat-resistant endospores and mitosporic fungal species [24,25]. Liquid spawn is an alternative method that has recently attracted much attention for large-scale cultivation. Liquid spawn obtained by submerging fermentation techniques can yield mycelial biomass and a more uniform mycelial biomass in a shorter period [26]. Sawdust and rice bran substrates are suitable for Polyporale species cultivation for linear growth and spawn production [12,27,28]. However, in contrast with earlier studies, oyster mushrooms cultivated on sawdust substrates have more recently been shown to demonstrate a low yield and biological efficacy [29]. This could be attributed to the low protein content of lignocellulosic materials in sawdust, which is insufficient for commercial mushroom production [30].

Meanwhile, other studies have demonstrated that the use of rice straw as a sawdust substrate can produce high yield efficacy [31,32]. According to Chang et al. (1995), sawdust and rice bran substrates are suitable for Polyporale mycelial growth and fruiting body development [13]. MBL can be produced using agro-waste substrates, sawdust, rice straw, husk, bran, and other lignocellulosic materials [16,33]. This approach is low-cost, eco-friendly, and free of hazardous reagents and chemicals [33].

*F. fraxinea* (ASTI 17001) is the best strain for MBL production and is a species of white-rot fungus with high medicinal value. It grows on different tree logs and all types of forest and agro-waste materials. In addition, *F. fraxinea* produces versatile lignin-degrading enzymes, including laccase, lignin peroxide, and manganese peroxide. The fruiting body is thick, with a leathery texture and bright pigmentation. Mycelium-based leather production does not involve mixing mycelia with a cultivation substrate. In addition, bio-foam and construction materials, such as mycelial bricks, have been developed using fungal mycelium and lignocellulosic substrates.

Nevertheless, MBL material development depends on the type of fungus and substrate (solid/semi-solid), growing conditions, and processing of the material [34]. A previous study by Bae et al. (2021) demonstrated that the fungal species, substrate type, and environmental conditions affect the physical strength of the mycelium [18]. The chitin polymer (N-acetyl glucosamine) is the main basal layer of the fungal cell wall and is a pivotal factor in the development of the mycelium [35], acting as a moisture barrier to prevent leaching and smooth the surface. Plasticizers are liquids, have low molecular weights with low volatility, are soft mycelium/biocomposite materials, and increase flexibility [36].

Fungal/mushroom cell-wall components are not constant; atmospheric air and high temperatures can assist in the degradation of cell-wall proteins and phenolic compounds. Plasticizing, coating, crosslinking, and other approaches are the most conspicuous. Protein- and lipid-based plasticizers reduce brittleness and hardness, while simultaneously increasing the elongation and ductility of bio-based and polymer materials [37]. In contrast, polysaccharides and polyphenolic compounds may increase the stiffness of mycelium-based materials [38]. However, plasticizers should be natural and biodegradable with low toxicity and good compatibility with mycelial biomaterials. The most commonly used plasticizing agents are glycerol, polyethylene glycol, PEG 400, mono/di/oligosaccharides, lipids, and lipid derivatives [39]. Glycerol is the most prevalent hydrophobic film-making technique and decreases the intermolecular attraction between polymers and biomaterials [38]. These substances simultaneously reduce hardness and density while increasing the MBL flexibility. In addition, they are hygroscopic and leach when in contact with water. 

The mycelium-based material was heat-killed after growth. The purpose of this research was to develop sustainable leather and characterize the physicochemical and mechanical properties of MBLs made with agro-waste types of lignocellulosic reinforcement substrate material containing the Polyporales species, *F. fraxinea*. The thermal stability and water content-angle coefficient of the MBLs and different plasticizing and hotpress treatments in all aspects were compared with those of the control. Hotpress processing can substantially increase the tensile strength. Water contact-angle tests revealed the hygroscopic and hydrophobic nature of the MBL. With average weight loss in the temperature range of 300–500 °C, the control samples and PEG hotpressed MBLs showed high heat stability (Figure 7a–d). Cartabia et al. (2021) reported that fungal cell wall degradation and weight loss depend on cultivation conditions and cell-wall chemistry [40]. These experimental results show that MBLs can partially fulfill the requirements of commercial leather and have the potential to replace bovine leather. The methodology used to evaluate the suitability of plasticizing and crosslinking approaches proved effective for the fabrication of MBLs. To our knowledge, this is the first study that reports the chemical, physical, and mechanical characteristics of MBL from *F. fraxinea*.

Albers et al. (2020) demonstrated the water-swelling behavior of PEG-based polyurethane networks [41]. The differences in the cell-wall chemical composition of plasticizers and the hotpress treatment can explain their diverse hydrodynamic behavior, especially regarding moisture uptake. Furthermore, the water contact angle is dependent on the surface topography [42,43]. Crosslinking and coating may increase the water content. Appels et al. (2020) observed that the mycelium density could increase after 2–8% glycerol treatment [44]. Many studies have revealed that the tensile resistance of mycelium-based biomaterials is more influenced by binder failure [45,46]. The mechanical properties of the developed MBLs are important to measure because mycelia are leather materials, which can respond differently to stress and strain.

Moreover, with and without hotpress treatment, MBL samples are structurally inhomogeneous in cell-wall component distribution, orientation, and thickness, which adds a stochastic character to the mechanical response. The measured parameters, such as elongation percentage, stress, and strain, showed significant differences among all samples, taking into account different plasticizing, coating, crosslinking, and hotpress treatments. In general, untreated mycelial samples are stiffer than MBLs and have less elongation at fracture [43]. In addition, the coating reduces the water absorption and smooths the surface. Plasticizing and crosslinking are essential steps for altering tensile and elongation strengths. Gennadios (2002) reported that pure corn zein-treated film samples had comparable flexibility to animal leather [47]. Furthermore, the biofabrication of mycelial mushroom mats is promising for the use of MBL in industrial applications using agro-waste. However, considering the plasticizing effect of cell-wall polysaccharides already observed above and the possibility of developing MBLs through different approaches, such as crosslinking and coating, these properties can be easily tuned.

Furthermore, plasticized mycelial samples maintained better hydrodynamic, elongation, and tensile strengths than the control sample. The tensile strength and Young’s modulus were recorded in 60–120 °C hotpress-treated samples. The mycelial mat treated with over-layered 20% PEG, such as cheesecloth, showed high tensile strength and Young’s modulus scores at 120 °C. The physical properties of mycelium-based materials and bovine leather are listed in Table 3. Simultaneously, the tensile strength increased with a decrease in elongation. The chemical composition of MBLs treated with 20% PEG was recorded as carbohydrate, chitin, and protein from the FTIR band range of 3000 to 3650 cm^−1^. Among these, the carbohydrate content was the highest. The fungal mycelium cell wall comprises multiple layers that vary in chemical composition [48]. In addition, other cell-wall components, such as mono-sugars (glucose), deoxy-sugars (rhamnose), and amino sugars (glucosamine), were detected. Glucosamine is the building unit of polysaccharide chitin and chitosan synthesis by fungi and mushrooms [49]. Zhou et al. (2013) reported polysaccharides derived from mushrooms composed of glucose, rhamnose, mannose, and monosaccharides [50].

The percentage of fungal cell-wall components varied, depending on the species, cultivation of the substrate, and the manufacturing process. The SEM investigation of 20% PEG hotpress-treated MPLs from the top, middle, and bottom views showed a smooth, less porous, rigid, and compact mycelial architecture. The images exhibited an interconnected network and surface features of the MBLs and control samples. Antinori et al. (2020) reported that surface-poured mycelium-based materials are used in drug delivery, tissue engineering, and enzyme immobilization [43]. The SEM images of the bottom surfaces of the MBLs were compact, whereas the upper and middle surfaces were uneven and fragmented. Furthermore, the 20%-PEG-hotpress MBL surface was smooth and resembled leather. The pure mycelial mats developed from wood decay fungus were hairy (upper), whereas the bottom side was smoother and more compact [40]. The EDX spectrum was used to study the adsorbent composition of the MBL samples. The control and treated samples exhibited strong intense peaks of C and O and weak elemental signals of S, K, Mg, and Ca in the MBLs.

## 5. Conclusions

Fungal/mushroom-based materials are highly relevant in the discussion of environmental issues pertaining to the development of sustainable biomaterials. Recently, mushroom-based leather production has attracted significant attention because of its low production cost, ecofriendly and nonchemical processes, and contribution to valorized agro-waste substrates. Strain selection and growth optimization are critical for MBL production. We screened 14 Polyporale species for MBL production. Among these, *F. fraxinea* is a suitable strain for leather production. Simultaneously, high-density mycelia were obtained by solid-state fermentation of 30–40 d. Physicochemical and mechanical characteristics and morphological images revealed that *F. fraxinea* is a suitable strain for MBL production. In addition, cheesecloth-over-layered 20% PEG with hotpress mycelium exhibited high tensile strength and Young’s modulus scores. Hybridization and strain improvement are crucial for competitive mushroom-based leather production. Fungal/mushroom-based leather has been suggested as a future competitor for animal and synthetic leather because it offers a sustainable alternative to small leather goods. However, mushroom-based leather production is limited to a few companies globally. The production process is improving rapidly and may come to supply high-quality leather materials at a reasonably low cost in the future.

## Figures and Tables

**Figure 1 jof-08-00317-f001:**
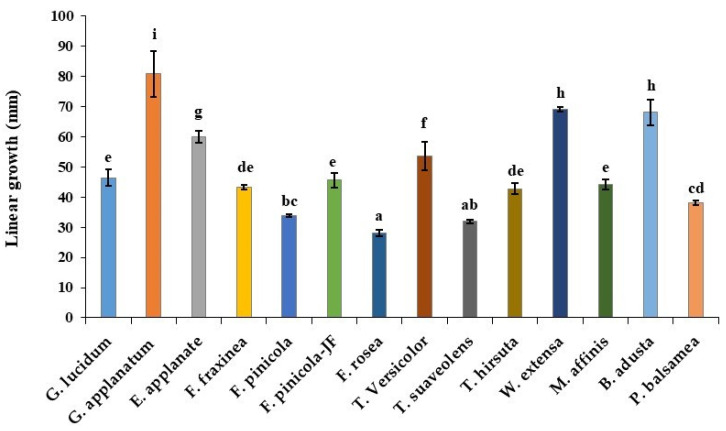
Linear growth measurement of the Polyporales species. Each value is the mean of three individual triplicate experiments (±SD), and species with the same letter do not differ significantly by the Duncan test, (*p* < 0.05). (*p* < 0.05)—the lower case letter: *p*-value less than 0.05; mm—millimeter.

**Figure 2 jof-08-00317-f002:**
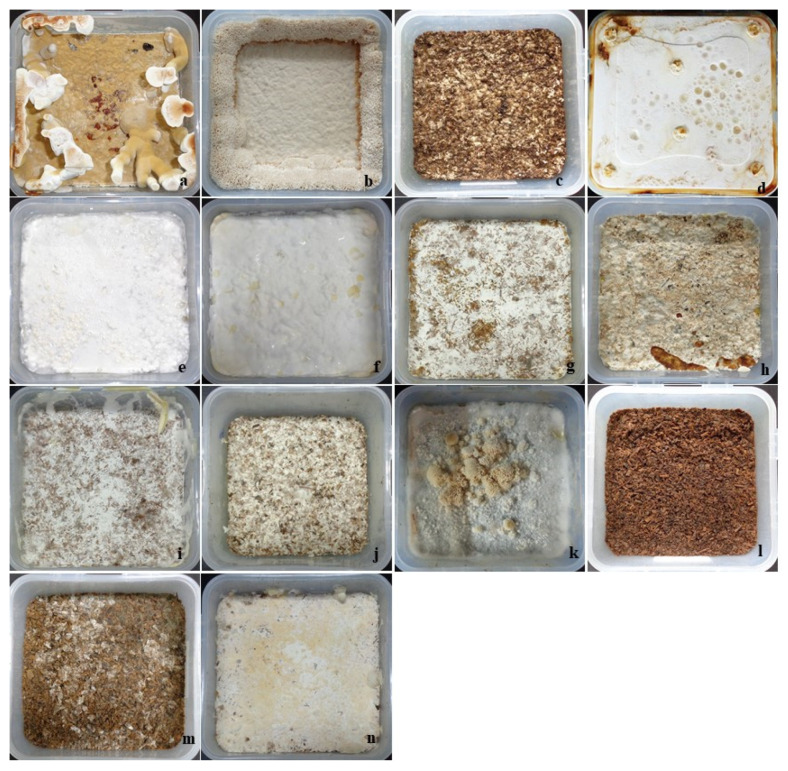
Topography of the mycelial mat generated from the Polyporales species. (**a**) *Ganoderma lucidum*, (**b**) *G. applanatum*, (**c**) *Elfvingia applanate*, (**d**) *Fomitella fraxinea*, (**e**) *Formitopsis pinicola*-JF, (**f**) *F. pinicola*-KCTC, (**g**) *F. rosea*, (**h**) *Trametes versicolor*, (**i**) *T. hirsuta*, (**j**) *T. suaveolens*, (**k**) *Wolfiporia extensa*, (**l**) *Microporus affinis*, (**m**) *Bjerkandera adusta*, (**n**) *Postia balsamea*.

**Figure 3 jof-08-00317-f003:**
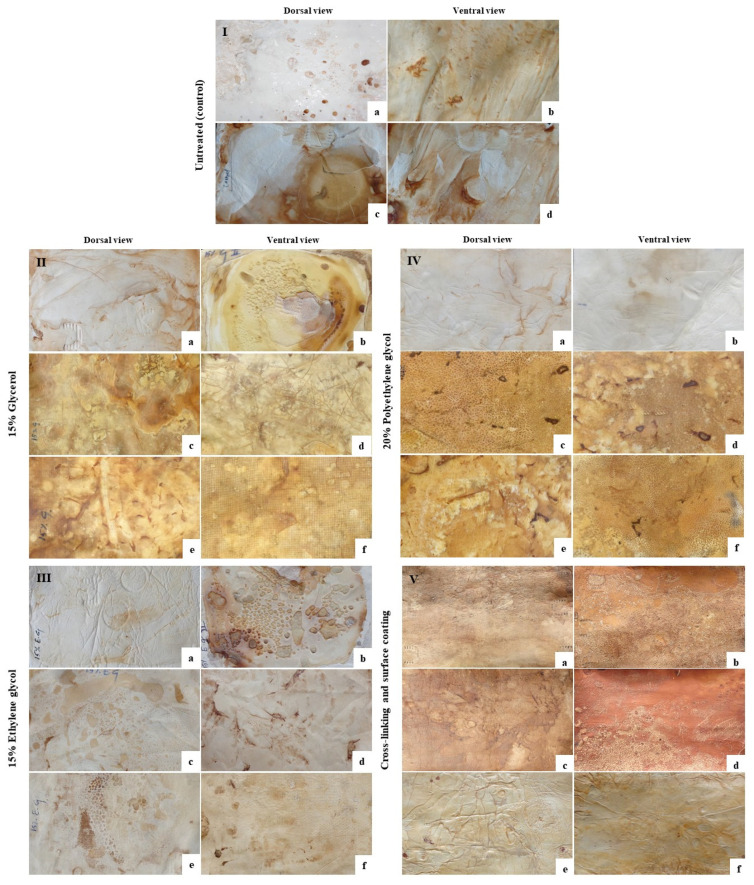
Surface view of the plasticized and hotpressed MBL obtained from *F. fraxinea*. (**I**) Untreated control (**a**,**b**) fresh mycelium, (**c**,**d**) hotpressed; (**II**) 15% glycerol-treated (**a**,**b**) without hotpress, (**c**,**d**) hotpressed, (**e**,**f**) cheesecloth overlaid with hotpress; (**III**) 15% ethylene-glycol-treated (**a**,**b**) without hotpress (**c**,**d**) hotpressed, (**e**,**f**) cheesecloth overlaid with hotpress; (**IV**) 20%-PEG-treated (**a**,**b**) without hotpress; (**c**,**d**) hotpressed; (**e**,**f**) cheesecloth overlaid with hotpress; (**V**) crosslinked and surface-coated (**a**,**b**) 15% EG + 5% tannic acid, (**c**,**d**) 15% EG + 10% tannic acid, (**g**,**h**) 15% EG + 20% coated with corn zein.

**Figure 4 jof-08-00317-f004:**
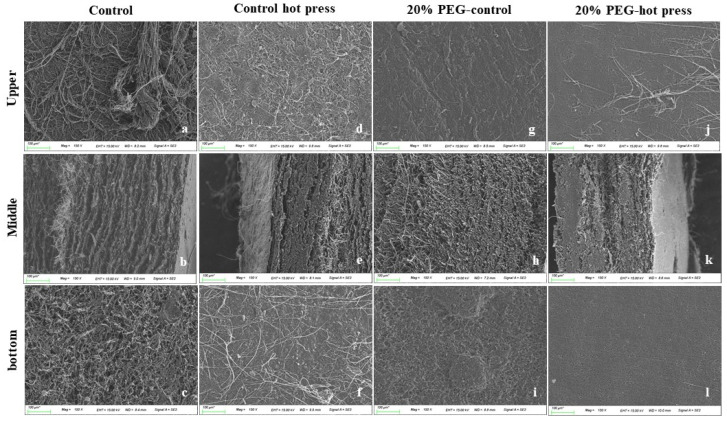
Scanning electron microscopy (SEM) images of MBLs. (**a**–**c**) control; (**d**–**f**) control hotpress; (**g**–**i**) 20% PEG control; (**j**–**l**) 20% PEG hotpress samples. Scale bars are marked with 100 µm.

**Figure 5 jof-08-00317-f005:**
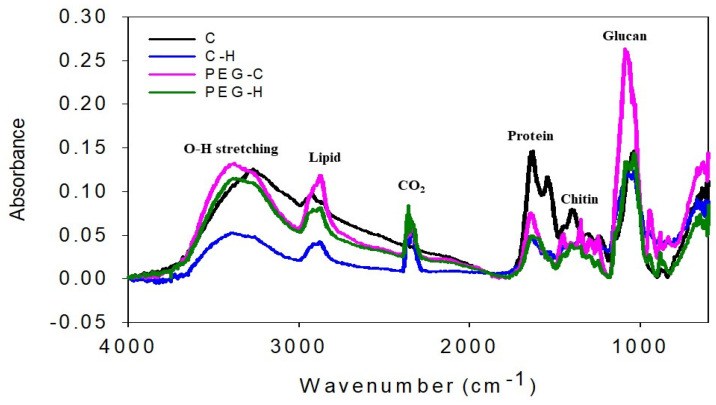
FTIR spectra of MBLs, showing those of the control samples and pretreated MBL samples. The main absorption bands indicating the mycelial cell-wall components are the asymmetric and symmetric vibration O-H stretching, demonstrating a high glucan content. The major absorption bands and their principal groups are labeled. As PEG-C samples exhibit a chemical effect, a difference between the plasticized MBLs, the hotpress control, and treated samples can be observed.

**Figure 6 jof-08-00317-f006:**
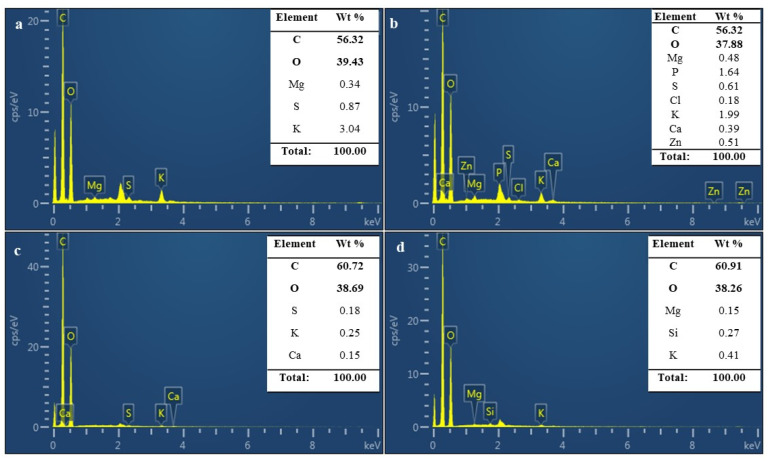
EDX analysis of MBLs. (**a**) control, (**b**) control hotpress, (**c**) 20%-PEG-control, and (**d**) 20%-PEG-hotpress samples.

**Figure 7 jof-08-00317-f007:**
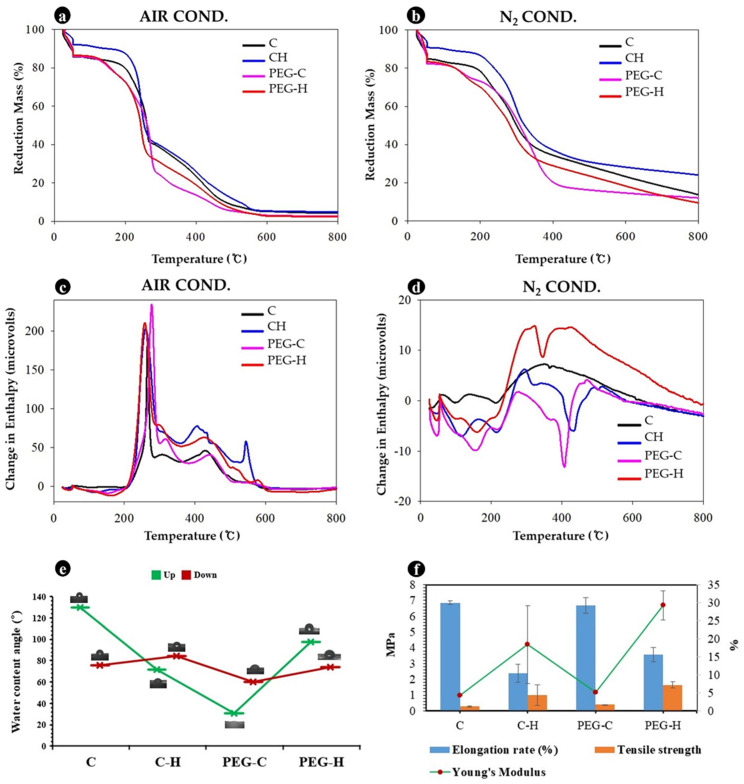
Chemical properties and mechanical properties of MBLs. (**a**,**b**) Thermal gravimetric analysis (TGA) assuming (**a**) air or (**b**) nitrogen atmosphere, with (**a**) slight weight increase observed (N_2_ content reducing point increased, and the slight weight increase is in probably slow decomposition of the MBLs); (**c**,**d**) differential scanning calorimetry analysis (DTA); (**e**) water contact angle; (**f**) mechanical properties overview.

**Table 1 jof-08-00317-t001:** Colony morphology, growth rate, and characteristics of Polyporales species.

Strain	Culture Number	Radial Growth (mm in 4 d) *	Average Linear Growth (mm in 16 d) ^#^	Fresh Weight ^$^	Dry Weight ^$^	Initial Moisture Content (%) ^$^	Mycelial Characteristics *
*G. lucidum*	JF 17-01	20.33 ± 0.68 ^f^	46.46 ± 2.85 ^e^	7.17 ± 0.34 ^de^	0.98 ± 0.04 ^abc^	94.74 ± 0.51 ^b^	Whitest brown, compact, and highly dense with brown patches in the middle
*G. applanatum*	KMCC 02967	41.42 ± 0.13 ^i^	80.88 ± 7.52 ^i^	6.45 ± 0.61 ^bcd^	0.98 ± 0.03 ^abc^	94.06 ± 1.12 ^b^	Yellowish-white, compact, and rapid growth Mycelium and highly dense
*Elfvingia applanate*	JF 26-01	11.75 ± 0.77 ^a^	59.90 ± 2.00 ^g^	5.54 ± 1.27 ^b^	1.26 ± 0.37 ^bc^	86.28 ± 8.14 ^a^	Creamy white, compact, highly dense, exudate on surface, and wrinkled mycelium
*Fomitella fraxinea*	ASTI 17001	18.17 ± 0.79 ^de^	42.77 ± 0.78 ^de^	4.07 ± 0.34 ^a^	0.81 ± 0.02 ^a^	95.15 ± 0.39 ^b^	White, highly dense, and slow growth
*Fomitopsis pinicola*	JF 79-01	17.42 ± 0.47 ^d^	33.85 ± 0.50 ^bc^	5.54 ± 0.41 ^b^	1.12 ± 0.07 ^abc^	90.10 ± 1.21 ^ab^	Creamy white, thin, fluffy in corner, and dense
*F. pinicola*	KCTC 6208	18.75 ± 0.67d ^ef^	45.63 ± 2.43 ^e^	7.60 ± 0.66 ^ef^	1.02 ± 0.09 ^abc^	94.57 ± 1.09 ^b^	Yellowish-white, thin, and dense
*F. rosea*	KCTC 26226	13.58 ± 0.52 ^b^	28.10 ± 1.13 ^a^	5.74 ± 0.09 ^bc^	1.12 ± 0.01 ^abc^	90.87 ± 0.56 ^ab^	White to pinkish-white, cottony, fluffy in corner, and thin
*Tramets versicolor*	JF 52-01	25.33 ± 0.56 ^g^	53.67 ± 4.74 ^f^	10.92 ± 0.28 ^h^	1.33 ± 0.51 ^c^	93.38 ± 4.75 ^b^	Yellowish-white and dense
*T. suaveolens*	KCTC 26205	19.42 ± 0.47 ^ef^	31.90 ± 0.67 ^ab^	8.38 ± 0.27 ^fg^	1.15 ± 0.02 ^abc^	93.51 ± 0.35 ^b^	Creamy white and dense
*T. hirsute*	KCTC 26200	26.17 ± 0.68 ^g^	42.81 ± 1.75 ^de^	11.33 ± 0.17 ^h^	1.36 ± 0.02 ^c^	93.25 ± 0.11 ^b^	Creamy white and dense, and colonies are powdery
*Wolfiporia extensa*	JF 46-01	32.17 ± 0.68 ^h^	69.16 ± 0.83 ^h^	4.33 ± 0.76 ^a^	0.91 ± 0.08^ab^	92.05 ± 3.96 ^ab^	Brownish-white, fluffy, upright mycelial, thin, and moderate growth
*Microporus affinis*	JF 47-01	15.17 ± 0.47 ^c^	44.19 ± 1.80 ^e^	6.80 ± 0.81 ^cde^	1.21 ± 0.01 ^abc^	90.49 ± 2.70 ^ab^	White-pink and dense
*Bjerkandera adusta*	JF 78-01	41.00 ± 0.39 ^i^	68.17 ± 4.26 ^h^	5.68 ± 0.10 ^bc^	1.19 ± 0.24 ^abc^	89.07 ± 5.07 ^ab^	Creamy white, thin, floccose texture, and rapid growth
*Postia balsamea*	JF 80-01	24.83 ± 1.81 ^g^	38.10 ± 0.69 ^cd^	9.32 ± 0.38 ^g^	1.27 ± 0.05 ^bc^	92.73 ± 0.69 ^ab^	White, dense, in vitro teleomorph formation was observed, and exudate drops

* Radial growth and mycelial morphology characteristics on YMPA media. ^#^ Linear growth on sawdust substrate. ^$^ YMPB liquid culture. d—days; mm—millimeter; %—percentage, the lower case letter: *p*-value less than 0.05.

**Table 2 jof-08-00317-t002:** Mechanical properties of MBL from *F. fraxinea* (G—glycerol, EG—ethylene glycol, PEG—polyethylene glycol (GCO, EGCO, and PEGCO with ‘CO’ denoting cheesecloth-over-layered MBLs). Each value represents the mean (±SD), and species with the same letter do not differ significantly by the Duncan test (*p* < 0.05). MBLs—Mycelium-Based Leather Samples; MPa—Mega Pascal; %—percentage, °C—degree celsius.

No.	MBL Samples (MBLs)	Elongation Rate (%)	Tensile Strength (MPa)	Strain	Young’s Modulus (MPa)
**Without hotpress**					
1	Control	4.59 ± 1.29 ^a^	1.40 ± 0.22 ^ab^	1.03 ± 0.01 ^ab^	1.37 ± 0.20 ^ab^
2	15% G	69.74 ± 5.33 ^j^	1.60 ± 0.07 ^ab^	1.35 ± 0.03 ^j^	1.18 ± 0.05 ^ab^
3	15% EG	37.88 ± 5.28 ^gh^	7.00 ± 1.74 ^h^	1.19 ± 0.03 ^gh^	5.91 ± 1.61 ^gh^
4	20% PEG	29.29 ± 2.15 ^ef^	1.74 ± 0.05 ^ab^	1.49 ± 0.03 ^k^	1.16 ± 0.06 ^ab^
**Hotpress (60 °C)**					
5	Control	25.41 ± 3.29 ^de^	2.65 ± 0.26 ^ab^	1.13 ± 0.02 ^def^	2.48 ± 0.20 ^bc^
6	15% G	58.86 ± 5.19 ^i^	4.92 ± 1.05 ^efg^	1.29 ± 0.03 ^i^	3.79 ± 0.74 ^de^
7	15% EG	33.56 ± 5.96 ^fg^	5.09 ± 1.11 ^efg^	1.17 ± 0.03 ^fg^	4.35 ± 0.90 ^ef^
8	15% G + 5% tannic acid	22.64 ± 1.15 ^bcde^	3.13 ± 0.62 ^bcd^	1.11 ± 0.01 ^de^	2.81 ± 0.54 ^cd^
9	15% EG + 5% tannic acid	23.55 ± 3.17 ^cde^	2.58 ± 0.06 ^ab^	1.07 ± 0.06 ^bcd^	2.41 ± 0.18 ^abc^
10	15% EG + 20% corn zein	43.73 ± 7.82 ^h^	1.35 ± 0.26 ^a^	1.22 ± 0.02 ^h^	1.10 ± 0.20 ^a^
11	20% PEG	18.88 ± 6.33 ^bcd^	5.06 ± 0.78 ^efg^	1.09 ± 0.03 ^d^	4.62 ± 0.67 ^efg^
**Hotpress (120 °C)**					
12	Control	3.62 ± 1.00 ^a^	3.99 ± 0.45 ^cde^	1.02 ± 0.01 ^a^	3.94 ± 0.46 ^de^
13	15% G	20.32 ± 1.90 ^bcde^	4.54 ± 0.23 ^def^	1.11 ± 0.02 ^de^	4.08 ± 0.18 ^de^
14	15% GCO	16.73 ± 5.65 ^bc^	6.16 ± 0.80 ^gh^	1.08 ± 0.03 ^cd^	5.68 ± 0.66 ^fgh^
15	15% EG	29.92 ± 1.67 ^ef^	5.74 ± 0.09 ^fgh^	1.15 ± 0.01 ^efg^	4.99 ± 0.11 ^efg^
16	15% EGCO	20.9 ± 1.82b ^cd^	6.28 ± 0.89 ^gh^	1.10 ± 0.01 ^de^	5.60 ± 0.97 ^fgh^
17	20% PEG	15.89 ± 2.02 ^b^	7.21 ± 0.93 ^hi^	1.08 ± 0.01 ^bcd^	6.69 ± 0.67 ^h^
18	20% PEGCO	8.6 ± 0.44 ^a^	8.49 ± 0.90 ^i^	1.04 ± 0 ^abc^	8.14 ± 0.88 ^i^

Elongation rate—percentage of elongation, measured to capture the amount an MBL would plastically and elastically deform up to fracture. Tensile strength—measurement of the MBL’s ability to withstand longitudinal stress, expressed as the maximum stress that the MBL can withstand without breaking. Strain—value describing the relative deformation or change in the shape and size of MBLs under applied force. Young’s modulus—ratio of tensile strength (σ) and strain (ε).

**Table 3 jof-08-00317-t003:** Physical properties of mycelium-based materials and bovine leather (elongation, tensile strength, and Young’s Modulus).

Type	Elongation (%)	Strain	Tensile Strength (MPa)	Young’s Modulus (MPa)	Ref.
Fungal mycelium (S. c) ^#^	-	2.2	12.3	1048	[49]
Fungal mycelium (G. l) ^#^	14–33	-	0.8–1.1	-	[36]
Mycelium composite ^#^	-	-	0.1–0.2	66.14–71.77	[45]
Mycelium-based composite (T. m) ^#^	0.9	4.7	0.15	100	[51]
Mycelium leather *	22–35	-	8–11	-	[52]
Mycelium based leather *	4.59–58.86	1.03–1.49	1.40–8.49	1.37–8.14	Present study
Animal leather	<40	6–16	39.5	1–13	[53,54]

S. c. *Schizophyllum commune*, G. l. *Ganoderma lucidum*, T. *Trametes multicolor*, ^#^ mushroom-based materials, * mycelial leather.

## Data Availability

Not applicable.

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
