# Peer review of "Mycofabrication of Mycelium-Based Leather from Brown-Rot Fungi"

_jof, 2022, doi:10.3390/jof8030317_

Round 1
Reviewer 1 Report
In this study, the physicochemical and mechanical properties of mycelium-based leather (MBL) samples from Polyporales were investigated as a sustainable substitute for leather that can be produced from fungal mycelium. This study is very interesting and important for the further development of sustainable alternative products. The experiments are good-designed and the obtained results are convincible. Therefore, this manuscript can be recommended for publication at the Journal of Fungi after minor revisions.
In the Introduction part, application areas of mushroom products such as filtration, decontamination, vertical farming, clothing, accessories, etc. should be mentioned. Companies are known for their mushroom products are Mylo (USA), Coffins (Dutch), Ecovative (USA), MYCL (Indonesia), Brodo, Mogu, Mycoflex, etc. Meanwhile, the big brands like Stella McCartney, Adidas, Lululemon and Hermes also use partnerships with biotechnology startups Bolt Threads and MycoWorks, etc. and sell, handbags, flooring and sound-proofing acoustic panels. Perhaps these studies would be a good fit.
Mycofiltration for removal of contaminants from water DOIhttps://doi.org/10.1186/s13750-021-00232-0
Mycelium-composite panels for atmospheric particulate matter adsorption https://doi.org/10.1016/j.rinma.2021.100208
Vertical farming: materials 2019, 12(14), 2270; https://doi.org/10.3390/ma12142270
Line 98 Potato Dextrose Agar (PDA) please write out in full.
From which companies were yeast, malt, and peptone extract agar purchased?
Line 129 What does 6 d mean, is it meant 6 days here?
Line 132 It is not entirely comprehensible how the examinations were conducted. What does "under stationary conditions" mean? With which device (model, manufacturer's country, etc.) were shaking conditions performed? At what temperature?
Line 141 The mycelial-grown substrate was transferred to boxes (155 × 155 × 87 mm3, HPL822D, LOCK & LOCK, Korea). What material are these boxes made of?
Line 148 What material is this by cheesecloth, cotton or another material? What does the fungal mycelium on cheesecloth look like? The pictures here can help to better understand the fungal mycelium composite.
The abbreviations such as PEG, SEM, FTIR, etc. should be written out. Please check the complete manuscript for abbreviations and write them out fully.
Line 151: It is not very clear how mycelial harvesting and mycofabrication were done and should be described in more detail. The mycelial mats were drained (with what device and how exactly?), dried at room temperature (were mushroom mycelial mats hung or placed on a surface or perhaps stretched?), and uniformly rolled (with what device were they rolled and under what conditions?).
Line 163: Please specify manufacturer, model, etc. for standard bench manual presses. Under what conditions were cool-pressed mat dried for 48 h?
Could you specify sawdust and rice bran?
How was average linear growth calculated?
Fig. 2. The labels of a,b,c...n should be a little larger because they are difficult to read.
Fig. 3 Labels are a little too small.
How was hot-pressing performed, with which machine?
Line 311 How do authors know that coating reduces the water absorption and smoothens the surface? On what results from these statements are based?
Fig. 4 Scale bars are very small in pictures.
Line 436 Four different spawn types are mentioned here, but it is not said exactly which ones are meant.
Line 519 Which industrial applications are meant here?
Line 571 Which few companies are meant here?
Author Response
Answers to the reviewer’s comments
Reviewer 1
In this study, the physicochemical and mechanical properties of mycelium-based leather (MBL) samples from Polyporales were investigated as a sustainable substitute for leather that can be produced from fungal mycelium. This study is very interesting and important for the further development of sustainable alternative products. The experiments are good-designed and the obtained results are convincible. Therefore, this manuscript can be recommended for publication at the Journal of Fungi after minor revisions.
In the Introduction part, application areas of mushroom products such as filtration, decontamination, vertical farming, clothing, accessories, etc. should be mentioned. Companies are known for their mushroom products are Mylo (USA), Coffins (Dutch), Ecovative (USA), MYCL (Indonesia), Brodo, Mogu, Mycoflex, etc. Meanwhile, the big brands like Stella McCartney, Adidas, Lululemon and Hermes also use partnerships with biotechnology startups Bolt Threads and MycoWorks, etc. and sell, handbags, flooring and sound-proofing acoustic panels. Perhaps these studies would be a good fit.
Mycofiltration for removal of contaminants from water DOIhttps://doi.org/10.1186/s13750-021-00232-0
Mycelium-composite panels for atmospheric particulate matter adsorption https://doi.org/10.1016/j.rinma.2021.100208
Vertical farming: materials 2019, 12(14), 2270; https://doi.org/10.3390/ma12142270
Line 98 Potato Dextrose Agar (PDA) please write out in full.
Response: We have decode the appreciation
From which companies were yeast, malt, and peptone extract agar purchased?
Response: We have given the company name [BD, Difco, USA]
Line 129 What does 6 d mean, is it meant 6 days here?
Response: ‘d’ means days and we have updated in the manuscript.
Line 132 It is not entirely comprehensible how the examinations were conducted. What does "under stationary conditions" mean? With which device (model, manufacturer's country, etc.) were shaking conditions performed? At what temperature?
Response: we have updated the corrections
[The fragmented mycelia were transferred to identical conical flasks and incubated for 3 d stationary (unrocked) and for 4 d agitation (rocked, 150 rpm), in dark at 28 °C]
Line 141 The mycelial-grown substrate was transferred to boxes (155 × 155 × 87 mm3, HPL822D, LOCK & LOCK, Korea). What material are these boxes made of?
Response: The boxes are made with polypropylene and we have updated the corrections
Line 148 What material is this by cheesecloth, cotton or another material? What does the fungal mycelium on cheesecloth look like? The pictures here can help to better understand the fungal mycelium composite.
Response: We have updated the sentence as “cheesecloth (woven cotton gauze fabric) grade 10 (20 × 12 threads per square inch)”.
The abbreviations such as PEG, SEM, FTIR, etc. should be written out. Please check the complete manuscript for abbreviations and write them out fully.
Response: We have abbreviated in the beginning (materials & methods section) of the manuscript.
Line 151: It is not very clear how mycelial harvesting and mycofabrication were done and should be described in more detail. The mycelial mats were drained (with what device and how exactly?), dried at room temperature (were mushroom mycelial mats hung or placed on a surface or perhaps stretched?), and uniformly rolled (with what device were they rolled and under what conditions?).
Response: We have updated the corrections according to the reviewer suggestion
“The mycelial mats were drained (hanged with clip), dried at room temperature (drying fan, 48 hours with low speed), and rolled evenly (manual hand press roller, METLUCK)”.
Line 163: Please specify manufacturer, model, etc. for standard bench manual presses. Under what conditions were cool-pressed mat dried for 48 h?
Response: We have updated the corrections
Could you specify sawdust and rice bran?
Response: we have specified the sawdust in materials & methods section and result section.
We have used oak sawdust and rice bran substrate (8:2, ratio)
How was average linear growth calculated?
Response: We have given details about the linear growth measurement in materials & method section 2.3.
Fig. 2. The labels of a,b,c...n should be a little larger because they are difficult to read.
Response: We have enlarged and change the color of the image labels
Fig. 3 Labels are a little too small.
Response: We have enlarged the image labels
How was hot-pressing performed, with which machine?
Response: We have updated details in materials & methods section.
Line 311. How do authors know that coating reduces the water absorption and smoothens the surface? On what results from these statements are based?
Response: The data not presented in this manuscript.
Fig. 4 Scale bars are very small in pictures.
Response: We have enlarged the scale bars in Fig. 4.
Line 436 Four different spawn types are mentioned here, but it is not said exactly which ones are meant.
Response: Its was mentioned in the discussion section, we have used liquid spawn in the study. “Recently, four different spawn types have been used for large-scale mushroom cultivation [22, 23].”
Line 519 Which industrial applications are meant here?
Response: We have updated the correction.” Furthermore, the bio-fabrication of mycelial mushroom mats is promising for the use of MBL in leather industrial applications using agro-waste.”
Line 571 Which few companies are meant here?
Response: We have included the companies name in the discussion section (MycoWork, Bolt Threads, Desserto, Mycotech Lab, and Mycel)
Reviewer 2 Report
Article Mycofabrication of mycelium-based leather from brown-rot 2
fungi by Jegadeesh Raman, Da-Song Kim, Hyun-Seok Kim, Deuk-Sil, Oh, Hyun-Jae Shin contains interesting content that shows the results of a development that has the potential to create a new material that replaces the skin of animals in industrial production. This is relevant in an area of ​​high demand and changing environmental and humanitarian preferences, as well as in connection with the controversial, but still possible importance of declining livestock to maintain the climate.
The article is written in a simple understandable language, which makes it accessible, but is excessively promotional in structure and presentation of the material. In connection with these shortcomings, the work can be accepted, but after changes and quite significant ones.
The introduction section should be changed. Things done in the work should not be placed in the introduction - they should be placed in the results, discussion, conclusion (conclusions). In the introduction, it is necessary to highlight the goals and objectives, and move lines from 75 to 83 to the sections corresponding to them.
It is equally important to determine in the introduction which properties of the obtained polymers are decisive, how new materials will be compared with traditional ones, which are intended to replace them, according to these or other criteria.
Goals and objectives should be formulated in a dedicated paragraph.
In section 2.1. Strains applied in the study should indicate the cultivation vessels and their volumes, quantitative sampling, cultivation duration, method of sowing and storage, or provide a link.
Section 2.2. "The diameter of mycelial growth was measured (days 1 to 4), and the mycelial density was determined qualitatively, classified as very thick (++++), thick (+++), thin (++), and very thin (+)" should describe the thickness of the mycelium more explicitly. For example in mm. It is also not clear how ++ differs from + and so on. This should be explained.
Section 2.5. Box cultivation and substrate preparation (line 137) indicate weight or volume of substrate
Please specify the method of sample preparation, in particular sample drying in section 2.8. Physical and mechanical properties analysis.
It is not clear what is meant by "Cell wall chemistry" (line 178). Either describe how the samples were obtained or change the term.
In "Table 1. Colony morphology, growth rate, and characteristics of Polyporales species" the range of validity of the statistics should be indicated.
Figure 2 needs correction, the letters are hard to see, enlarge them please.
Figure 7 needs correction, the letters are hard to see, enlarge them please.
However, the PEG-C samples showed a more hydrophobic nature than PEG-H (line 393) - confirm or indicate on what the claim is based.
The manuscript contains a somewhat lengthy conclusion and two general phrases that would be correct if you add a discussion of perspective and complexity. However, it is better to post it in the discussion.
The manuscript contains high-quality meaningful material and, with little effort, will be of great use to readers and researchers in the field.
Author Response
We have revised the whole manuscript according to Reviewer 2's comments. Please find the pdf file attached.

Reviewer 3 Report
+ attached file
review for
Article 1
Mycofabrication of mycelium-based leather from brown-rot 2 fungi 3
by Jegadeesh Raman1,2, Da-Song Kim1, Hyun-Seok Kim3, Deuk-Sil, Oh3, Hyun-Jae Shin1,*
leather from fungi is a hot topic in science and business
many start up and companies are already running
---------------------------------------------------
example 1
Mushroom Leather Startup MycoWorks Lands $125 Million In Latest Funding Round
FASHIONALT MATERIALS
By Jill Ettinger Published on Jan 17, 2022 Last updated Jan 30, 2022
1
1
Share
3 Mins Read
Turning mushrooms into leather is proving to be big business for MycoWorks, the Bay Area startup that has produced vegan leather for a number of brands including luxury French fashion house Hermès. Now, MycoWorks has raised $125 million in its Series C funding round to scale.
The new funding for MycoWorks will go to scale its manufacturing, namely for its flagship vegan leather product called Reishi.
The company says its goal is to meet the demand for sustainable materials and goods. The funding will go to advance R&D efforts and technology. According to MycoWorks, it is inundated with requests from brands interested in using its materials.
Scaling production
The Series C funding builds on its Series B $45 million raise and $17 million in its Series A since launching in 2013.
According to MycoWorks, it produces a fabric different from its competitors. It says its biotechnology platform allows it to engineer the mycelium—the root structure of mushrooms—to grow into a made-to-specification luxury material.
Courtesy
“A lot is happening in this space,” MycoWorks CEO Matt Scullin told TechCrunch. “Mycelium is a tunable material, and a lot of folks are entering the space because they see opportunity for it. However, their main approach is taking fibers and embedding them in plastic, which results in a low-quality material like ‘pleather.’”
That made choosing the right investors critical, according to Scullin. Prime Movers Lab, one of the investors shares expertise in biotechnology and scaling manufacturing—two of MycoWorks biggest needs at the moment.
“What MycoWorks has achieved with its Fine Mycelium platform is not just a breakthrough, it is a revolution for industries that are ripe for change,” David Siminoff, general partner at Prime Movers Lab, said in a statement. “This opportunity is massive and we believe that unrivaled product quality combined with a proprietary scalable manufacturing process has MycoWorks poised to serve as the backbone of the new materials revolution.”
Mushroom leather goes mainstream
While partnerships like the Hermès deal have positioned MycoWorks as a luxury leather producer, the company’s aim is mass production, though, allowing for a range of products and price points. The company says the new funding is key to scaling.
Part of that will happen via a new production plant in South Carolina, expected to be operational within 12 months. It should be capable of producing several million square feet of mycelium per year, the company says. It already operates a west coast plant outside of San Francisco in Emeryville, California, where it recently produced 10,000 trays of the product it calls Fine Mycelium.
Courtesy
Disrupting leather is big business. By the company’s account, more than $150 billion worth of traditional leather is sold every year. An animal-based material, leather brings both ethical and sustainability concerns.
The alternative leather market, particularly mushroom leather, is growing. MycoWorks’ chief competitor, Bolt Threads, just saw its vegan mushroom leather in the new Mercedes-Benz concept EV. It’s also been used in designs by British fashion designer Stella McCartney. During Fashion Week last September, McCartney debuted a limited-edition handbag made with the vegan mushroom leather.
------------------------------------------------
example 2
Mycotech: The Indonesian Startup Making Mushroom Leather Inspired By Tempeh
ALT MATERIALSFASHIONVEGAN
By Sally Ho Published on Feb 27, 2021 Last updated Feb 26, 2021
676
73
Shares
3 Mins Read
Head over the Bandung, the city set in the middle of volcanoes and tea plantations in Indonesia, and you’ll find a startup working on turning waste into sustainable animal-free leather. Called Mycotech Lab, the company was inspired by tempeh, the traditional Indonesian food made from fermented soybeans, and came up with its own technology to grow its ethical and carbon-friendly mycelium-based materials.
Many are now shunning traditional leather made from cowhide, which is a high-emissions and resource-intensive material, not to mention unethical. But there aren’t yet many alternatives to leather that really perform in terms of both sustainability, with petroleum-based plastic alternatives still remaining the lesser of two evils, and functionality, since many animal-free fabrics lack the strength, durability and hand feel that producers and consumers like about real leather.
While there’s a growing cohort of startups trying to change that with their novel material innovations, many of them like MycoWorks, Bolt Threads, Desserto and Fruitleather are based in Europe or the U.S., but there’s Mycotech Lab right here in Asia whose founding story is deeply rooted in its local culture.
Mycotech’s team experimented with different mushrooms since 2016.
It all started when the team thought about the process of making tempeh, the traditional Indonesian protein made from fermenting soybeans with fungi, which has now become a huge hit in the West amidst the vegan surge, but has existed for centuries in Javanese cuisine and has featured in many delicacies across Asia for years.
Mycotech Lab decided to experiment with the fermentation process used to make tempeh to make a new fabric out of the complex root structure of mushrooms, otherwise known as mycelium. It was a lengthy trial-and-error process that kicked off in 2016, but “finally, we found one mushroom with a mycelium that can be made into binding material,” said Erlambang Ajidarma, head of research at the startup, in conversation with Reuters.
Mylea, the mycelium leather alternative developed by Mycotech.
The final product, developed with fungus grown on sawdust that then gets scraped off and dried and cut into different sizes, is Mylea, a fibrous but tough material that acts just like the real thing. It’s waterproof, pliable, durable, and most importantly, is far more sustainable than existing plastic-based synthetic leathers or carbon-intensive real leather made from hide.
Mycotech also uses natural dye extracted from roots, leaves and food waste in the region to colour their leather alternative, which again is a process that is far less polluting than traditional tanning processes used for real cowhide that leaves behind solid and liquid waste that contains chromium and other hazardous compounds.
Since its inception, Mycotech has managed to grow its client base with no marketing budget because the demand for sustainable alternatives has grown alongside awareness of the damaging effects of animal-based materials in the fashion industry.
Products made with Mycotech’s leather alternative.
One of their recent partnerships includes local sneaker brand Bro.do, who now plans to expand into other markets like Japan in response to popular demand. They’ve made bags, watch straps and accessories out of Mylea too.
In a recent interview with e27, the startup’s co-founder and CEO Adi Reza Nugroho revealed that clients have been “happy with the result” of their mycelium leather, and have attracted enough order volumes to stay busy until 2027.
All images courtesy of Mycotech Lab.
+ many many others
------------------------------------
only one mention in your text
- Mycotech Lab. Mylea technical data sheet.pdf. https://mycl.bio/storage/app/me-758 dia/mylea/Mylea%20Technical%20Data%20Sheet.pdf Accessed 29 Nov 2021
authors, in your introduction, please develop a long paragraph presenting the state-of-the-art of myco leather in the world
+ the way you introduce yourself in this (new aspects, novelty etc)
---------------------------------------
The most significant effects were observed after treatment with 20% polyethylene glycol, which resulted in an increase in the Young’s modulus and tensile strength.
Where your PEG is coming from? source=chemistry?
adding 20% of chemical in a ‘natural’ end product sounds strange
alternative, natural, origin for PEG???
OR alternative, natural, origin for a product ‘equivalent’ to PEG???

Author Response
We have revised the whole manuscript according to Reviewer 3's comments. Please find the pdf file attached.

Round 2
Reviewer 2 Report
Dear Authors,
Your manuscript has been improved, but I still recommend that you set aside the last paragraph of the introduction for the formulation of goals and objectives. There are no links in this paragraph. Also carefully look at the text, for example: line 676
20% PEG - molecular weight and polymer characteristics not specified.
In general, the manuscript has been improved and can be accepted.
Author Response
Dear Reviewer:
I have updated the introduction part and the PEG details in the Materials and method section.
Thank you for your kind suggestion.
Best regards,
Hyun-Jae Shin